# Risk factors for maternal mortality among 1.9 million women in nine empowered action group states in India: secondary analysis of Annual Health Survey data

Geneviève Horwood,[1] Charles Opondo [ID],[1] Saswati Sanyal Choudhury,[2] Anjali Rani,[3] Manisha Nair [ID] [1]

[1]Nuffield Department of Population Health, University of Oxford, Oxford, UK
[2]Department of Obstetrics and Gynaecology, Gauhati Medical College and Hospital, Guwahati, Assam, India
[3]Department of Obstetrics and Gynaecology, Banaras Hindu University Institute of Medical Sciences, Varanasi, Uttar Pradesh, India

**Correspondence to**
Dr Manisha Nair;
manisha.nair@npeu.ox.ac.uk

## ABSTRACT

**Objective** To examine the risk factors for pregnancy-related death in India's nine Empowered Action Group (EAG) states.

**Design** Secondary data analysis of the Indian Annual Health Survey (2010–2013).

**Setting** Nine states: Assam, Bihar, Chhattisgarh, Jharkhand, Madhya Pradesh, Odisha, Rajasthan, Uttar Pradesh and Uttarakhand.

**Participants** 1 989 396 pregnant women.

**Methods** Maternal mortality ratio (MMR), overall and for each state, with 95% CI was calculated. Stepwise multivariable logistic regression was used to investigate the association of risk factors with maternal mortality. Area under the receiver-operating characteristic (AUROC) curve was used to assess the prediction of the model.

**Outcome measures** MMR adjusted for survey design, adjusted OR (aOR) with 95% CI and C-statistic with 95% CI.

**Results** MMR calculated for the nine states was 383/100 000 live births (95% CI 346 to 423 per 100 000). Age exhibited a U-shaped association with maternal mortality. Not having a health scheme and belonging to a scheduled caste or scheduled tribe group were significant risk factors for maternal death with aOR of 2.72 (95% CI 2.41 to 3.07), 1.10 (95% CI 1.02 to 1.18) and 1.43 (95% CI 1.31 to 1.56), respectively. Socioeconomic status and rural residence were not associated with maternal mortality after adjusting for access to a healthcare facility. Complications of pregnancy and medical comorbidities were the strongest risk factors for maternal death (aOR 50.2, 95% CI 44.5 to 56.6). Together, the risk factors identified accounted for 89% (95% CI 0.887 to 0.894) of the AUROC.

**Conclusions** Maternal mortality in India's EAG states greatly exceeds the national average. The identified risk factors demonstrate the importance of improving the quality of pregnancy care. Notably, the study showed that the risk conferred by poor socioeconomic status could be mitigated by universal access to healthcare during pregnancy and childbirth.

### Strengths and limitations of this study

► This study identifies the risk factors for maternal mortality in nine socioeconomically disadvantaged states in India, a country with the second-highest number of maternal deaths, globally.

► The study examines >7000 maternal deaths in a population of >1.9 million pregnant women, which makes this the large population-based study examining a wide range of risk factors for pregnancy-related death in a vulnerable population in India where maternal death has been previously under-reported.

► The focus of this study was on high burden disadvantaged states, thus, the findings may not be generalisable to the rest of India.

► The survey design likely introduced non-systematic reporting bias leading to potential under-reporting of maternal deaths in early pregnancy and deaths due to indirect causes.

## INTRODUCTION

Despite advances in modern medicine, pregnancy and childbirth remain one of the leading causes of mortality worldwide for women of reproductive age.[1] The question of safe motherhood is not merely one of public health concern but has been widely recognised as an issue of social injustice.[2 3] According to the Sustainable Development Goals (SDGs), the global target is to reduce the maternal mortality ratio (MMR) to less than 70 per 100 000 live births by 2030 and to provide universal access to reproductive healthcare.[4]

India is responsible for the second-highest number of maternal deaths worldwide.[5] India's MMR has been steadily declining since the 1990s.[1 5 6] According to the Sample Registration System, a household survey conducted by the Indian government, the

MMR dropped from over 400 per 100 000 in the early 1990s, to 230 in 2008 and to 130 per 100 000 between 2014 and 2016.[5] In comparison, the global MMR in 2015 was estimated at 216 (95% CI 207 to 249).[6] However, this downward trend masks staggering within-country variations in maternal mortality. For example, in the southern state of Kerala, in India, the MMR was reported to be as low as 61 per 100 000 live births in 2013 whereas in the northern state of Bihar it was 208 per 100 000 live births.[5] This inequality in the burden of maternal deaths between northern and southern states is reflected by other socio-economic indicators such as poverty and education[7] which could influence access to care during pregnancy and postpartum.[8]

The government of India launched the National Rural Health Mission (NRHM) in 2005, to improve access to good quality healthcare services for socioeconomically disadvantaged population concentrating its efforts on nine Empowered Action Group (EAG) States.[9 10] These are Assam, Bihar, Chhattisgarh, Jharkhand, Madhya Pradesh, Odisha, Rajasthan, Uttar Pradesh and Uttarakhand. The EAG states and Assam together account for about 50% of the total population, 61% of the total births, 71% of the infant deaths and 72% of the under-5 deaths in India.[10] Under NRHM, multiple maternal health programmes were developed. Examples include the Janani Suraksha Yojana (JSY), a cash-incentive scheme,[11] whose objective is to promote institutional deliveries, and introduction of a large cadre of community health workers known as Accredited Social Health Activists (ASHAs). These health schemes are generally aimed at impoverished or marginalised women to facilitate access to healthcare.

Despite these efforts, little is known about the population-level risk factors that influence maternal death in the vulnerable population in these states. The objective of this study was to examine the risk factors for pregnancy-related death in India's nine EAG states using a large-scale population-based dataset.

## METHODS

### Study design
This study is a secondary data analysis of the Annual Health Survey (AHS), a government household survey conducted between 2010 and 2013 in the nine EAG states.

### Data source
The AHS is a large household survey of over 4 million households. The survey was conducted every year between 2010 and 2013 in the nine EAG states. Compared with the Demographic and Household surveys, AHS collected more detailed information on events and outcomes related to women's reproductive health and pregnancy occurring between 1 January 2007 and 31 December 2009.[10] The survey used a stratified simple random sampling strategy. It recorded information regarding demographic and socioeconomic characteristics, reproductive and sexual health information and maternal death. The maternal

---

> **Box 1 Definitions related to maternal death**
>
> **Maternal death**
> Death of a woman while pregnant or within 42 days of termination of pregnancy, irrespective of the duration and site of the pregnancy, from any cause related to or aggravated by the pregnancy or its management, but not from accidental or incidental causes.[5]
>
> **Maternal mortality ratio**
> Number of maternal deaths per 100 000 livebirths.[5]

---

death questionnaire gathered information from a member of the household in which a woman had died in pregnancy or postpartum during the reference period.[10] The questionnaire focused on identified delays to care and factors contributing to death, but the methodology was not comparable to the conventional verbal autopsy. The information on 'delay' in the dataset were mainly categorised into 'delay on the part of the woman or the family in recognising the danger signs or seriousness of the complication', 'delay in accessing healthcare' or 'delay in receiving appropriate healthcare'.

### Study sample
The study included 1 991 915 women reporting a pregnancy within the defined survey time frame and 7444 pregnancy-related deaths as reported by a member of the same household. Women were excluded if their age was outside the reproductive window. While the WHO considers the reproductive age to be 15–49 years, pregnancies occurring at age 13 and above were considered plausible and were included in the analysis. Women with missing survey weights or missing pregnancy outcomes were excluded. In the mortality data set, deaths occurring outside the time frame included in the definition of maternal death (see box 1) were excluded. While abortion-related deaths were recorded, the survey did not ask household members to identify factors associated with death in the case of women dying after an abortion. Therefore, all women, whether survived or died (including abortion-related deaths) were included in calculating the MMRs, but due to the lack of information, women who had an abortion in the reported index pregnancy (whether survived or died) were excluded from the risk factor analysis. Online supplementary figure S1 depicts the study sample derivation.

### Variable choice and extraction
A systematic search of the literature was undertaken by interrogating the MEDLINE, Embase and Global Health databases as well as the WHO's list of publications, to identify potential risk factors for maternal death in India. Using the literature search, hypothesised predictors of maternal mortality specific to our study setting were divided into broad categories of distal, intermediate and proximal factors guided by the WHO's conceptual framework of the social determinants of health.[12] The theoretical framework is depicted in online supplementary figure S2.

**Table 1** Baseline characteristics of the women included in the study population*

| | Surviving women | Maternal deaths |
|---|---|---|
| N | 1982398 | 6998 |
| Age (years)† | 26.0 (23–30) | 26.0 (22–32) |
| State | | |
| Assam | 168 040 (5%) | 812 (7.1%) |
| Bihar | 353 043 (18.5%) | 1247 (18.6%) |
| Chhattisgarh | 106 416 (4.1%) | 335 (3.5%) |
| Jharkhand | 162 987 (5.3%) | 647 (4.8%) |
| Madhya Pradesh | 233 964 (12.0%) | 742 (11.2%) |
| Odisha | 121 777 (4.7%) | 649 (6.5%) |
| Rajasthan | 150 668 (9.1%) | 529 (8.9%) |
| Uttar Pradesh | 538 160 (39.9%) | 1778 (38.6%) |
| Uttarakhand | 147 343 (1.5%) | 259 (0.8%) |
| Region of residence | | |
| Rural | 1 690 156 (82.8%) | 6118 (85.5%) |
| Urban | 292 242 (17.2%) | 738 (13.1%) |
| Missing‡ | 0 | 142 (1.5%) |
| Religion | | |
| Buddhist | 956 (<0.01%) | 2 (<0.01%) |
| Christian | 26 109 (1.1%) | 114 (1.2%) |
| Hindu | 1 632 888 (81.4%) | 5813 (83.0%) |
| Jain | 2074 (0.1%) | 4 (<0.01%) |
| Muslim | 292 640 (16.4%) | 903 (13.9%) |
| Sikh | 4713 (0.2%) | 14 (0.2%) |
| Other | 21 206 (0.5%) | 83 (0.6%) |
| Missing‡ | 1812 (0.1%) | 65 (1.1 %) |
| Social group | | |
| SC | 386 169 (20.6%) | 1533 (23.3%) |
| ST | 244 057 (10.5%) | 1082 (12.2%) |
| Others | 1 350 426 (68.7%) | 4318 (63.4%) |
| Missing‡ | 1746 (0.1%) | 65 (1.1%) |
| Has health scheme | | |
| No | 1 826 156 (92.7%) | 6552 (93.9%) |
| Yes | 15 6242 (7.3%) | 446 (6.1%) |
| Wealth index quintile | | |
| Poorest | 397 828 (20.1%) | 1492 (21.9%) |
| 2 | 397 338 (21.8%) | 1588 (23.9%) |
| 3 | 397 600 (20.4%) | 1400 (20.6%) |
| 4 | 395 890 (18.7%) | 1269 (17.5%) |
| Wealthiest | 393 742 (18.3%) | 1249 (16.1%) |
| Accessing a health facility | | |
| Yes | 1 066 136 (53.1%) | 2579 (36.1%) |
| No | 798 811 (40.1%) | 4322 (63.0%) |
| Missing‡ | 117 451 (6.01%) | 97 (0.9%) |
| Pregnancy complication or comorbidity | | |
| No | 1 460 887 (73.0%) | 535 (6.6%) |
| Yes | 521 511 (27.0%) | 6463 (93.4%) |

*Characteristics are presented as unweighted counts and survey-weighted frequencies.
†Age is presented as the median with IQR.
‡Missing reported here includes women with abortion-related deaths who were not included in the final model for risk factor analysis.
SC, scheduled caste; ST, scheduled tribe.

This theoretical framework guided the choice of variables used in the statistical analysis. The mortality questionnaire was brief, so few of the variables identified by the literature review were available. Of the distal factors identified in the framework, those available were place of residence, wealth index, having a health scheme (defined as whether the family received benefits from JSY, government health insurance and other health and financial benefit schemes), religion and social group (categorised as 'Schedule caste (SC)', 'Schedule tribe (ST)' and 'Others'; SC and ST are considered socioeconomically disadvantaged groups). Of the intermediate factors, the only variable available was accessing a health facility. For the proximal factors, a binary variable was created distinguishing between women with any complication of pregnancy or medical comorbidity and women with neither. Comorbidities included hypertensive disorders of pregnancy, sepsis, bleeding, jaundice, obstructed or prolonged labour, other pregnancy complications and other medical conditions. Women surviving pregnancy were classified as accessing a health facility if they delivered in a health facility. Women dying in pregnancy were classified as accessing a health facility if they died in a health facility. This classification was based on the presumption that both active labour and acute complications eventually leading to death were comparably crucial events in pregnancy requiring urgent medical care. The questionnaire addressed to surviving women had gathered further information regarding access to care in pregnancy including antenatal care and birth attendant. Conversely, the maternal death questionnaire only included place of death. Both groups' interaction with the healthcare system was therefore solely comparable based on place of death or delivery.

## Statistical analysis

Frequency distributions of binary and categorical variables as well as measures of central tendency for continuous variables were reported. Baseline characteristics were assessed and contrasted between maternal deaths and all surviving women. Frequency distributions of maternal deaths by identified delays to care and by time period of death in relation to pregnancy were examined, stratified by registered and unregistered deaths (defined based on whether the death was registered with the local registrar for vital statistics within the administrative unit). The MMR (defined in box 1) with 95% CI was calculated in each state and for the region overall using survey-weighted counts of maternal deaths and live births.

The univariable association of each potential risk factor with maternal death was explored using simple logistic regression. Stepwise logistic regression was then used to build a multivariable model. The sequential addition of variables to the model was informed by the theoretical framework, where distal factors were added first, followed by intermediate and proximal factors. Statistical interaction between the social group and wealth quintiles was tested. To inform appropriate modelling of the effect of age, tests for departure from linearity and trend were conducted. If the effects of risk factors were found to be markedly attenuated after full

adjustment, they were explored further by assessing confounding of the association by distal, intermediate and proximal factors.

Observations were not expected to be independent within clusters; therefore, analyses were adjusted for clustering and sampling design. Wald test estimation-values were reported and a modified Hosmer-Lemeshow test, appropriate for clustered data, was used to assess model goodness of fit.[13] To assess the potential effect of clustering on the variance of the model estimates, SEs were compared between models using Taylor linearised estimates of the variance and models using robust SEs. In addition, point estimates and CIs were compared between models with adjustment for clustering and those without adjustment.

Patterns and amounts of missing data were described for each of the included variables in the final model. While missing information could have been related to other independent variables, it was not thought to be associated with the outcome, and therefore, the data were presumed to be missing at random. An indicator variable for missingness was created for each variable with missing data to explore reasons for missingness using logistic regression. Since the outcome variable was binary (died or survived), multiple imputation by chained equations was used as a method of choice for imputing missing data, and models containing imputed datasets were compared with the complete-case analysis model. As the variables included in our analysis had a very small proportion of missing data, the complete-case analysis was retained as our final model.

To assess the risk prediction of each of the groups of risk factors (distal, intermediate and proximal), the area under the receiver-operating characteristic (AUROC) curves for the models corresponding to the addition of each of these groups were compared.

All analyses were carried out using Stata V.14.2. Two-tailed p values of less than 0.05 were considered statistically significant.

### Patient and public involvement

Patient and public were not involved in this secondary analysis of anonymous survey data.

### RESULTS

Table 1 shows the baseline characteristics of women included in the study. The median age was 26 years and was the same for women surviving and women dying in pregnancy. In general, the households with maternal deaths were more likely to be from a rural area and to be from the two lower wealth quintiles, to belong to an SC or ST social group and to not have a health scheme. Of the women surviving pregnancy, 53.1% accessed a health facility compared with only 36.1% of women dying in pregnancy. In addition, 92.4% of the women dying in pregnancy suffered at least one complication of pregnancy, compared with 26.3% of women surviving pregnancy.

### Maternal mortality ratio

The overall MMR was 383 per 100 000 live births (95% CI 346 to 423 per 100 000). Two states had an MMR of over 500: Assam (552, 95% CI 507 to 600 per 100 000) and Odisha (534, 95% CI 490 to 581 per 100 000). The state with the lowest MMR was Uttarakhand accounting for 209 maternal deaths per 100 000 live births (95% CI 182 to 239 per 100 000). No state had an MMR below 200 (figure 1).

### Delays contributing to maternal death and time period of death

Figure 2 depicts maternal death according to the identified delay most contributing to the death. The delay most frequently cited was inappropriate quality of care at the health facility (55.1%). Lack of care-seeking was the second most important delay (18.1%), followed by difficulty in reaching the facility (15.3%). In the groups expressing a delay in care-seeking or in reaching the facility, over 80% had not registered the death. Conversely, in those expressing a delay in quality of care, 57% had registered the death. The majority of the deaths were

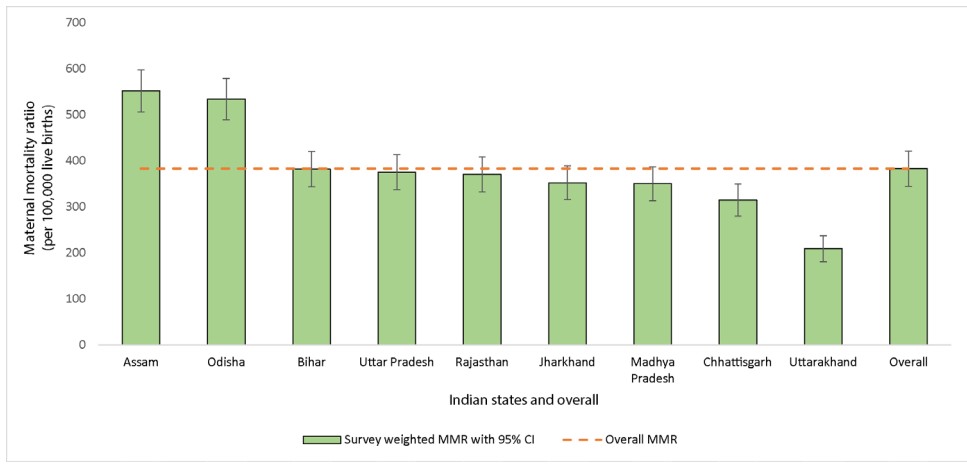

**Figure 1** Maternal mortality ratio (MMR) by state and overall. Source: Annual Health Survey, India.

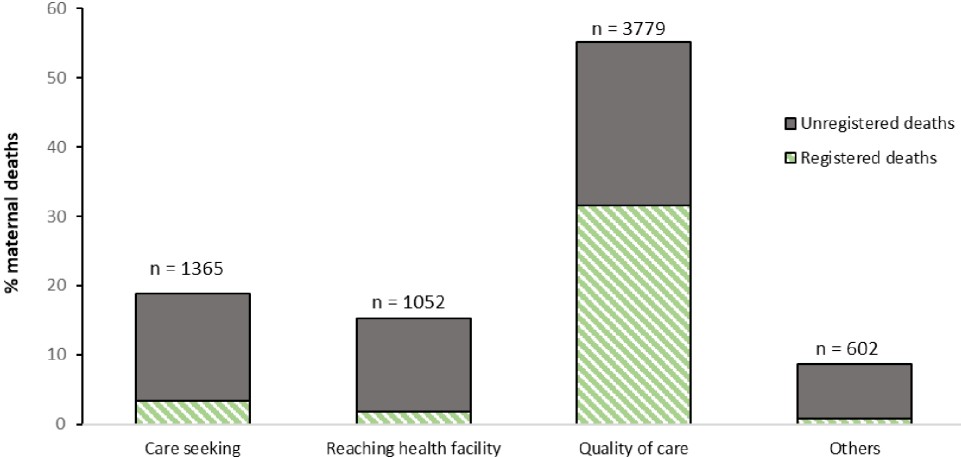

**Figure 2** Frequencies of maternal death according to the delay cited by household members as most contributing to the death. Note: maternal death frequencies are presented as survey-weighted frequencies. Unweighted counts of maternal deaths for each delay are presented over each bar. For each delay, maternal deaths are divided by the proportion of registered and unregistered deaths.

in the antenatal period (about 40%), followed by intra-partum and postpartum deaths. More than half of the deaths in each time period were unregistered (figure 3).

### Risk factors for maternal mortality

There was a departure from linearity in the association between age and maternal death and as such, age was coded as a categorical variable. Women between the ages of 25–29 years were least likely to die during pregnancy. The youngest age group, women between 13 and 19 years, was the group with the highest risk of death in pregnancy (adjusted OR (aOR) 3.66, 95% CI 3.27 to 4.10), followed by the eldest group of women of between 40 and 49 years of age (aOR 2.80, 95% CI 2.49 to 3.15). Age exhibited a strong U-shaped relationship with maternal death and this

relationship was largely unchanged after full adjustment for other risk factors (see online supplementary figure S3).

After adjustment for all distal factors, the effect size of the association between rural residence, social group, health scheme and maternal death were all mildly attenuated but remained significant. There was a trend of association across wealth quintile groups and maternal death. Associations between not having a health scheme, accessing a health facility or the presence of a complication with maternal death were all strengthened after adjustment for distal factors (table 2). However, attenuation of the associations of place of residence, wealth index and social group with maternal death occurred after adjustment for the presence of a complication or medical comorbidity

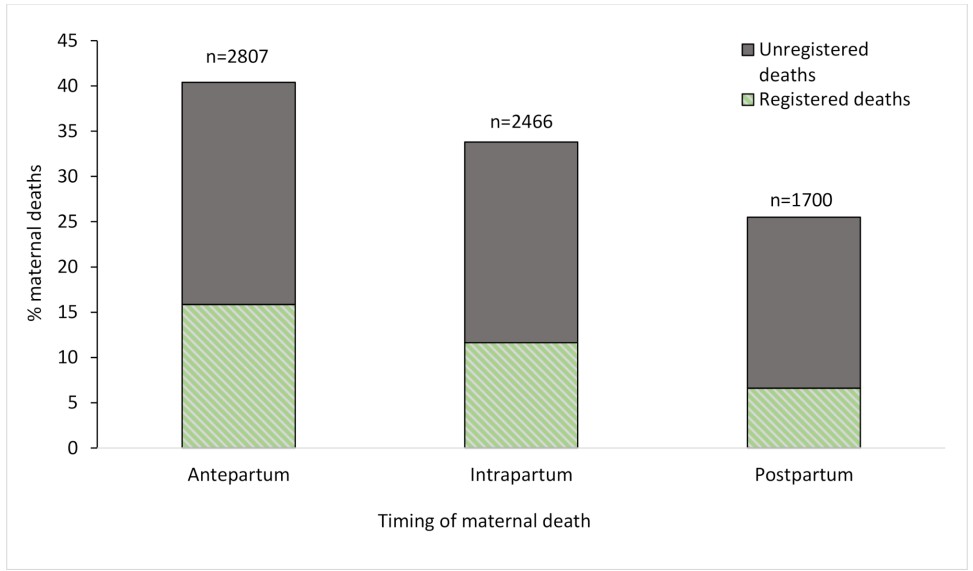

**Figure 3** Frequencies of maternal death according to the time period in relation to pregnancy. Note: maternal death frequencies are presented as survey-weighted frequencies. Unweighted counts of maternal death for each time period are presented over each bar. In each time period, maternal deaths are divided by the proportion of registered and unregistered deaths.

**Table 2** Risk factors for maternal mortality with progressive adjustment

| Variables | Unadjusted | | Adjusted for age | | Distal factors* | | Intermediate factors† | | Proximal factors‡ | |
|---|---|---|---|---|---|---|---|---|---|---|
| | OR | 95% CI | OR | 95% CI | OR | 95% CI | OR | 95% CI | OR | 95% CI |
| Age (y) | | | | | | | | | | |
| 13–19 | 4.23 | 3.80 to 4.73 | NA | – | 4.12 | 3.69 to 4.62 | 4.34 | 3.87 to 4.85 | 3.66 | 3.27 to 4.10 |
| 20–24 | 1.20 | 1.11 to 1.30 | NA | – | 1.20 | 1.11 to 1.30 | 1.25 | 1.15 to 1.35 | 1.16 | 1.07 to 1.26 |
| 25–29 | 1.00 | – | NA | – | 1.00 | – | 1.00 | – | 1.00 | – |
| 30–34 | 1.25 | 1.14 to 1.37 | NA | – | 1.25 | 1.14 to 1.37 | 1.20 | 1.10 to 1.32 | 1.19 | 1.08 to 1.31 |
| 35–39 | 2.06 | 1.86 to 2.27 | NA | – | 2.03 | 1.83 to 2.25 | 1.91 | 1.72 to 2.12 | 1.92 | 1.72 to 2.13 |
| 40–49 | 3.17 | 2.83 to 3.54 | NA | – | 3.13 | 2.79 to 3.51 | 2.91 | 2.60 to 3.27 | 2.80 | 2.49 to 3.15 |
| Distal factors | | | | | | | | | | |
| Place of residence | | | | | | | | | | |
| Urban | 1.00 | – | 1.00 | – | 1.00 | – | 1.00 | – | 1.00 | – |
| Rural | 1.34 | 1.23 to 1.47 | 1.28 | 1.17 to 1.40 | 1.23 | 1.12 to 1.34 | 1.05 | 0.95 to 1.15 | 0.97 | 0.88 to 1.07 |
| Religion | | | | | | | | | | |
| Hindu | 1.00 | – | 1.00 | – | 1.00 | – | 1.00 | – | 1.00 | – |
| Buddhist | 0.43 | 0.09 to 2.01 | 0.45 | 0.10 to 2.12 | 0.46 | 0.10 to 2.14 | 0.52 | 0.11 to 2.46 | 0.58 | 0.12 to 2.77 |
| Christian | 1.03 | 0.82 to 1.30 | 1.01 | 0.80 to 1.26 | 1.02 | 0.81 to 1.28 | 0.90 | 0.70 to 1.14 | 0.58 | 0.46 to 0.74 |
| Jain | 0.58 | 0.20 to 1.65 | 0.63 | 0.22 to 1.79 | 0.77 | 0.27 to 2.20 | 0.91 | 0.32 to 2.61 | 1.16 | 0.40 to 3.36 |
| Muslim | 0.83 | 0.76 to 0.91 | 0.79 | 0.73 to 0.86 | 0.84 | 0.77 to 0.92 | 0.76 | 0.69 to 0.83 | 0.64 | 0.59 to 0.70 |
| Sikh | 0.86 | 0.48 to 1.52 | 0.94 | 0.53 to 1.67 | 1.03 | 0.58 to 1.82 | 1.12 | 0.63 to 1.99 | 1.08 | 0.61 to 1.92 |
| Other | 1.03 | 0.78 to 1.37 | 0.96 | 0.72 to 1.27 | 0.88 | 0.66 to 1.18 | 0.76 | 0.56 to 1.02 | 0.72 | 0.54 to 0.97 |
| Social group | | | | | | | | | | |
| Other | 1.00 | – | 1.00 | – | 1.00 | – | 1.00 | – | 1.00 | – |
| Scheduled caste | 1.22 | 1.14 to 1.31 | 1.20 | 1.12 to 1.29 | 1.15 | 1.06 to 1.23 | 1.10 | 1.02 to 1.19 | 1.10 | 1.02 to 1.18 |
| Scheduled tribe | 1.24 | 1.14 to 1.34 | 1.19 | 1.10 to 1.29 | 1.15 | 1.05 to 1.26 | 1.11 | 1.02 to 1.21 | 1.43 | 1.31 to 1.56 |
| Wealth index | | | | | | | | | | |
| Wealthiest | 1.00 | – | 1.00 | – | 1.00 | – | 1.00 | – | 1.00 | – |
| Quintile 4 | 1.07 | 0.97 to 1.17 | 1.07 | 0.97 to 1.17 | 1.07 | 0.97 to 1.18 | 1.03 | 0.94 to 1.14 | 0.99 | 0.89 to 1.09 |
| Quintile 3 | 1.16 | 1.06 to 1.27 | 1.14 | 1.04 to 1.25 | 1.17 | 1.06 to 1.28 | 1.09 | 0.99 to 1.20 | 0.97 | 0.88 to 1.07 |
| Quintile 2 | 1.25 | 1.14 to 1.37 | 1.21 | 1.11 to 1.32 | 1.22 | 1.11 to 1.33 | 1.11 | 1.02 to 1.22 | 0.97 | 0.89 to 1.07 |
| Lowest quintile | 1.21 | 1.10 to 1.32 | 1.18 | 1.08 to 1.30 | 1.15 | 1.05 to 1.26 | 1.08 | 0.98 to 1.18 | 0.81 | 0.74 to 0.90 |
| Has health scheme | | | | | | | | | | |
| Yes | 1.00 | – | 1.00 | – | 1.00 | – | 1.00 | – | 1.00 | – |
| No | 1.24 | 1.10 to 1.39 | 1.26 | 1.12 to 1.41 | 1.36 | 1.21 to 1.53 | 1.33 | 1.18 to 1.50 | 2.72 | 2.41 to 3.07 |
| Intermediate factors | | | | | | | | | | |

Continued

**Table 2** Continued

| Variables | Unadjusted | | Adjusted for age | | Distal factors* | | Intermediate factors† | | Proximal factors‡ | |
|---|---|---|---|---|---|---|---|---|---|---|
| | OR | 95% CI | OR | 95% CI | OR | 95% CI | OR | 95% CI | OR | 95% CI |
| Accessed health facility | | | | | | | | | | |
| Yes | 1.00 | – | 1.00 | – | 1.00 | – | NA | – | 1.00 | – |
| No | 2.26 | 2.13 to 2.40 | 2.21 | 2.09 to 2.35 | 2.22 | 2.08 to 2.36 | NA | – | – | 2.93 |
| Proximal factors§ | | | | | | | | | | |
| No | 1.00 | – | 1.00 | – | 1.00 | – | 1.00 | – | 1.00 | – |
| Yes | 36.5 | 32.8 to 40.7 | 36.2 | 32.6 to 40.3 | 40.1 | 35.9 to 44.8 | 50.2 | 44.5 to 56.6 | NA | – |

*Further adjusted for distal factors.
†Further adjusted for accessing health facility.
‡Further adjusted for proximal factors.
§Pregnancy complication or medical comorbidity.
¶All analyses use logistic regression and are adjusted for clustering and survey design.

(table 2). Being of an SC or ST group continued to be significantly associated with maternal death. Not having a health scheme conferred significant risk (aOR 2.72, 95% CI 2.41 to 3.07) after adjustment for other factors.

After taking into account, the access to a health facility, the association between rural residence and maternal death was attenuated (aOR 1.05, 95% CI 0.95 to 1.15). In addition, the trend of association between wealth quintile groups and maternal death was no longer significant and there was no evidence of association between wealth index and dying in pregnancy. Online supplementary figures S4 and S5 depict the attenuation of the association between maternal mortality and both wealth and rural residence with sequential adjustment for other risk factors.

The magnitude of the association between proximal factors (pregnancy complications and medical comorbidities) and maternal death was further increased after adjustment for the distal and intermediate factors (aOR 50.2, 95% CI 44.5 to 56.6).

F-statistics adjusted for survey design as a measure of contribution to the model fit showed that the variables accounting for the greatest amount of variation in the data were proximal factors, accessing a health facility, not having a health scheme, and age. Model diagnostics showed no evidence of collinearity between dependent variables or model misspecification.

In assessing clustering of the data, SEs were compared between models with survey-weighted data using Taylor linearisation variance estimates and proportion-weighted data using robust SEs. These were found to be equivalent. Point estimates and SEs were similar between models with unweighted data and survey-weighted data. These findings suggested that clustering effects were minimal in the study population. Further, the findings were comparable in imputed models and complete-case analysis with a marginal widening of the 95% CIs. See online supplementary tables S1–S3 in the supplementary file.

Figure 4 shows the AUROC curve for each group of risk factors. Together, the distal and intermediate factors

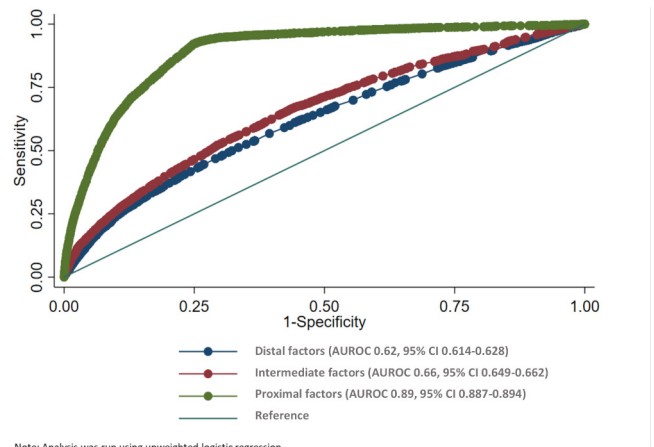

Distal factors (AUROC 0.62, 95% CI 0.614-0.628)
Intermediate factors (AUROC 0.66, 95% CI 0.649-0.662)
Proximal factors (AUROC 0.89, 95% CI 0.887-0.894)
Reference

Note: Analysis was run using unweighted logistic regression.

**Figure 4** Area under the receiver-operating characteristic (AUROC) curve for distal, intermediate and proximal factors. Note: analysis was run using unweighted logistic regression.

accounted for 66% (95% CI 0.649 to 0.662) of the variance in the data. With the addition of the proximal factors, this increased to 89% (95% CI 0.887 to 0.894).

## DISCUSSION

Our study showed that the MMR in the nine states was much higher than the corresponding WHO estimates for India's overall MMR for a similar time period. Lack of quality care in the health facilities was perceived as the factor most contributing to the maternal deaths by family members of deceased women. Proximal factors, which encompassed any pregnancy complication or medical comorbidity, were the strongest predictors of women dying in pregnancy followed by lack of access to the health facility. More importantly, the study showed that the effects of living rurally and being poor could be mitigated by improving access to the health facility. Other risk factors identified were younger or older maternal age, belonging to a scheduled tribe or caste social group, and not being enrolled in a health scheme.

The WHO reported MMR for India in 2010 was 208/100 000 live births[1 6] compared with 383/100 000 live births calculated in our study population for a similar period (2007–2009). This discrepancy reflects the high burden of maternal mortality in India's nine socioeconomically disadvantaged states. Moreover, in our study, more than half of the maternal deaths had not been previously recorded. Efforts to create a national electronic database for maternal death surveillance in India began in 2013,[14] but the progress is still limited, mainly due to lack of resources.[15 16]

Our study identified two key modifiable health systems determinants of maternal mortality in the nine disadvantaged states—access to the health facility and quality of care. Research in many low/middle-income countries (LMICs) consistently show that living in rural areas and being poor are associated with decreased odds of childbirth in the health facility, antenatal care and skilled attendance at birth.[17 18] Our study demonstrated that the effects of wealth and living rurally were mediated by a woman's ability to access care and do not exert a direct effect on maternal mortality. This is an important finding in the context of India and other LMICs in terms of universal access to antenatal, delivery and postnatal care. Access to appropriate good quality care could mitigate the effects of socioeconomic inequalities know to be associated with a high burden of maternal mortality.

The AUROC curve analysis provides evidence that proximal factors (pregnancy complications and medical co-morbidities) were the strongest predictors of maternal mortality in this population. Further, the association between proximal factors and maternal death was strengthened after accounting for other factors showing that these complications remain the most important risk factors for maternal death, thus reinforcing the importance of high-quality antenatal care, availability of basic emergency obstetrical care and postnatal care.

Nonetheless, in our study population, poor quality of care was reported as the factor most contributing to the maternal death in 55% of the deaths.

In 2005, the Indian government implemented the NRHM to address gaps in maternal healthcare. Improving the access to and quality of care, particularly for rural areas in high focus states, were the main objective of this initiative.[9] Since 2005, India has seen some improvements. Institutional deliveries increased from 39% in 2005 to 79% in 2013. Complete antenatal care coverage increased from 37% to 51% and postnatal care increased from 27% to 36%.[19] However, studies show that the programme has had little or no effect on clinical outcomes. For example, despite the increase in health facility deliveries, there has been no demonstrated decline in maternal mortality.[11 20]

Studies have repeatedly shown that quality of care is lacking in public health facilities, including non-availability of essential medicines such as uterotonics and antibiotics, and lack of facilities for caesarean section and blood transfusion.[21] Further, government reports on the state of public facilities in these nine states have reported lack of toilet facilities, unreliable electricity and unsatisfactory cleanliness,[22] and lack of basic measures of obstetrical care such as adequate hand hygiene.[23] Moreover, there have been reports of verbal and physical abuse of staff towards patients.[8 22]

Conforming with the findings of other studies,[24 25] age and maternal death exhibited a strong U-shaped association in which the youngest (women aged 13–19 years) and the oldest age group (women aged 40–49 years) had the highest risk of dying. India's National Family Health Survey showed that of women aged 20–49 years, 27% had been married before the age of 15 years and 58% were married before the legal age of marriage of 18 years. In addition, 30% of Indian women had given birth by age 19 years.[19] While age itself is not a modifiable risk factor, childbearing age should be regarded as one. Women should be able to exert control over when they choose to become pregnant. Policies supporting universal access to contraception, family planning counselling, safe abortion and promoting women's empowerment need continued attention in India.

This is a large population-based study conducted using a survey with minimal sampling bias and included a sample size of almost 2 million women and over 7000 recorded maternal deaths. The AHS database had two advantages over other household surveys in India: (1) it had more detailed information about a range of potential risk factors that we set out to examine in this study and (2) it allowed us to focus our analysis on socioeconomically disadvantaged states with a high burden of maternal deaths. We were able to study a population where maternal deaths are often unreported and where vital statistics and health-related data are lacking. One limitation is that the AHS was conducted in 2010–2013, and it is possible that the risk factors identified in our study may have changed over the years. However, they are unlikely to have changed significantly, considering that these nine states continue to have a higher maternal mortality rate than the rest of India. The risk factors

included in the regression model were informed by both established theory in the field of maternal health and a current review of the literature specific to the study setting, and the analysis addressed distal, intermediate and proximal causes of maternal death crucial to the understanding of women's health dynamics in the most vulnerable population in India's EAG states.

We acknowledge that information collected about women dying in pregnancy was limited and did not include several factors identified as important by the literature review such as education, paid employment, parity, age at first pregnancy, distance to the health facility or having a skilled birth attendant. However, a majority of these factors would have been covered by the 'wealth index' and 'access to health facility'. The survey design is also likely to have introduced a non-systematic reporting bias. In keeping with previous literature, under-reporting of early pregnancy deaths, particularly those arising from complications of abortion or deaths prior to detecting pregnancy, and of maternal deaths arising from indirect causes that might have been reported as deaths due to medical causes (ignoring the pregnancy state) could have led to an underestimation of maternal deaths.[26]

Maternal mortality in India's EAG states greatly exceeds the national average and is far from the target set out by the SDGs. Analysis of the results in the context of India's healthcare system suggests that targeted interventions to improve maternal outcomes in this population would include improved in-hospital care, improved access to healthcare services and guaranteed skilled personnel and basic emergency obstetrical care in every maternity centre. Reducing maternal mortality in India's socioeconomically disadvantaged states is achievable but will require continued accelerated national efforts to recognise the importance of women's lives and invest in sustainable health systems founded on principles of accountability and evidence-based medicine.

**Contributors** GH reviewed the literature, conducted the analysis and wrote the first draft of the paper; CO supervised the data analysis, interpretation and discussion of the results, and edited the paper; SSC helped in acquiring and interpreting the data, and edited the paper; AR edited the paper; MN developed the concept for the study, supervised the data analysis, interpretation and discussion of the results, and edited the paper.

**Funding** The study was funded by a Medical Research Council Career Development Award to Manisha Nair (Grant Ref: MR/P022030/1). The funder had no role in the study design, data analysis, data interpretation or writing of the manuscript. All authors, had full access to all of the data in the study and can take responsibility for the integrity of the data and the accuracy of the data analysis.

**Competing interests** None declared.

**Patient and public involvement** Patients and/or the public were not involved in the design, or conduct, or reporting, or dissemination plans of this research.

**Patient consent for publication** Not required.

**Ethics approval** Ethics approval was not required since this was a secondary analysis of anonymous survey data.

**Provenance and peer review** Not commissioned; externally peer reviewed.

**Data availability statement** Data are available in a public, open access repository. The anonymised data is freely available through the Indian Government's Data Sharing Portal.

**ORCID iDs**
Charles Opondo http://orcid.org/0000-0001-8155-4117
Manisha Nair http://orcid.org/0000-0003-0660-5054

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
