## [Reviewer comments · BMJ Open]

ARTICLE DETAILS

TITLE (PROVISIONAL)	Risk factors for maternal mortality among 1.9 million women in nine Empowered Action Group states in India: secondary analysis of Annual Health Survey data
AUTHORS	Horwood, Geneviève; Opondo, Charles; Choudhury, Saswati; Rani, Anjali; Nair, Manisha

VERSION 1 – REVIEW

REVIEWER	Elizabeth Anderson International Center for Research on Women United States
REVIEW RETURNED	09-Apr-2020

GENERAL COMMENTS	Thanks for the opportunity to review a useful contribution. A few comments below. Abstract: AUC is not really an outcome measure, it is a measure of model fit of your outcome measure; c-statistic is a generally more accepted name for this measure. Quite a few typos, starting in the abstract. Strengths and limitations: the second bullet is more of a statement of facts than a strength; consider rewording to explain why such a large sample size is valuable. Introduction: in the second paragraph, it would be useful to include the global MMR as a reference point. Can you provide a little more background on the EAG states and the specialized health efforts there? Methods: please define health scheme as it is not universally used. You mention sensitivity analyses via MICE but please include more detail of your approach. Results: in the first paragraph, what do you mean by "accessed a health facility"? Accessed ANC, gave birth at a facility, or something else? The maternal mortality ratio paragraph doesn't seem to add anything more than what is already available in Figure 1. Consider revising. Also, consider separating the "overall" columns in Figure 1 or at least putting them at one end or the other, rather than in between the various states. Can you include a qualitative interpretation of the results of your sensitivity analyses? They do not seem to be mentioned again in text after the methods.
---

	Discussion: please choose one name for the area under the curve and use it throughout; there are several inconsistent names used. Page 15, line 51, I would advise moving away from a description like "demonstrates convincingly". Perhaps "provides evidence" is more appropriate.
--	--

REVIEWER	Monica Saucedo Inserm U1153, Obstetrical, Perinatal and Pediatric Epidemiology Research Team. France
REVIEW RETURNED	27-Apr-2020

GENERAL COMMENTS	Thanks for give the opportunity to review this interesting paper. This study estimate the maternal mortality ratio and identifies the risk factors for maternal mortality in nine socioeconomically disadvantaged states in India, a country with the highest RMM, using information from the Indian Annual Health Survey (2010-2013). The study population included more than 7000 maternal deaths among >1.9 million pregnant women. The results should be of interest to readers of BMJ Open in particular from LIMIC. The article is well written and the results are clearly presented I have few comments and suggestions for authors : In the abstract, please give de 95% CI for MMR, as well as in the text P5, L13, it will be useful for the readers to know the recorded information regarding maternal deaths in the AHS questionnaire, in the supplementary material. Also in the methods, please give more details about documentation of maternal deaths, the data from AHS for maternal deaths is comparable to a "verbal autopsy" ? Are data about time of death available? During pregnancy, during labor and delivery or in the puerperium? P5L40: It is confusing the study population figure (S1), the exclusion from abortion. In the figure it is stated "abortion related deaths " n=94 517. Please give the n for survivors and for pregnancy related deaths. P6L49, identified delays to care appear only in the statistical analysis, could you please describe them early in the methods indicating how they were assessed. P9 L3, In the groups expressing a delay in care seeking or in reaching the facility, over 80% had not registered the death. Is this mean that there wasn't a death certificate for them? Where the deaths are registered in general?
---

REVIEWER	Santiago Garcia-Tizon Larroca Hospital General Universitario Gregorio Marañon de Madrid. Servicio de Ginecología y Obstetricia. Madrid. Spain
REVIEW RETURNED	Madrid. Spain 10-May-2020

GENERAL COMMENTS	First of all I would like to thank you for the opportunity to review this piece of work from the authors as a great effort to address and analyze relevant risk factors related to maternal mortality in such a big sample of patients. This is an outstanding study and I hope I can give you some ideas that might boost your manuscript for publication if you consider them suitable for your research. 1-Article summary/Abstract/Strengths and limitations of this study
--

	In the last point you mention that “the survey design likely introduced...under-reporting of maternal deaths in early pregnancy and deaths due to indirect causes” I see that this is also described in the DISCUSSION section (page 17, line 52) but the reasons why this happened are not explained. Is it possible for you to report this issue? Is it documented elsewhere? If it is not possible you should say it. It is positive that you mentioned it though. 2-Introduction Page 4, line 4. You mention that the burden of maternal deaths is reflected by other socio-economic indicators such as poverty and education. Is it possible for you to report more bibliography related to this ? Maybe other social determinants of health such as family income, poverty index of the maternal country of origin or human development index of the maternal country of origin that have been related to this complication? 3-Methods/Variable choice and extraction Page 6, line 16. You use a classification based on a theoretical framework where you divide factors into 3 groups: Distal, intermediate and proximal. Is this a classification created by you? Is it the WHO theoretical framework? I see the WHO report “The Social Determinants of Maternal Health. Adapted from WHO (2011) Closing the Gap: Policy into Practice on Social Determinants of Health. Discussion for the World Conference on Social Determinants of Health. Geneva: World Health Organization” (https://apps.who.int/iris/bitstream/handle/10665/44731/9789241502405_eng.pdf?sequence=1&isAllowed=y) where intermediate factors are described but I'm not sure you are referring to this. You should report where you got this way of classification from or if you just made it yourself, which is perfectly ok. Anyway I understand it is a very clear way to present it also with figure S2 so as a reader I appreciate it. 4-Tables S1, S2 and S3 You describe proximal factors as YES or NO but you don't specify causes of maternal death or the main medical conditions related to maternal comorbidity. I don't see it in the RESULTS section either. I understand that this is not the purpose of the study but I wonder if it will be published in the future or it was already done previously as a very valuable information for clinicians like me. For instance the main clinical cause of maternal death is haemorrhage throughout the world. Numbers of this complication are quite different between populations regarding social determinants. Congratulations again for your work again, please take my comments more as suggestions than corrections.
--	--

VERSION 1 – AUTHOR RESPONSE

Response: We have revised the statement as advised.

Reviewer(s)' Comments to Author:

Reviewer: 1

Reviewer Name

Elizabeth Anderson

Institution and Country

International Center for Research on Women United States

Please state any competing interests or state 'None declared':

None declared

Please leave your comments for the authors below

Thanks for the opportunity to review a useful contribution. A few comments below.

Response: We thank the reviewer for the comments and suggestions.

Abstract: AUC is not really an outcome measure, it is a measure of model fit of your outcome measure; c-statistic is a generally more accepted name for this measure.

Quite a few typos, starting in the abstract.

Response: Thank you for the comment and we apologise for any typing errors. We have revised the abstract as suggested.

Strengths and limitations: the second bullet is more of a statement of facts than a strength; consider rewording to explain why such a large sample size is valuable.

Response: We have reworded the point, as advised, to reflect the strength of the paper rather than merely including a statement which the reviewer rightly pointed out.

Introduction: in the second paragraph, it would be useful to include the global MMR as a reference point.

Can you provide a little more background on the EAG states and the specialized health efforts there?

Response: As suggested by the reviewer, we have revised the introduction section to include the global MMR as a reference point, and have also added a few sentences to describe the background of the EAG states, and the specialised health efforts in these states.

Methods: please define health scheme as it is not universally used.

Response: Thank you for the comment. We have included a sentence in the revised draft to explain what was meant by 'health scheme'.

You mention sensitivity analyses via MICE but please include more detail of your approach.

Response: We have added the following information in the revised draft –

“While missing information could have been related to other independent variables, it was not thought to be associated with the outcome, and therefore, the data were presumed to be missing at random. An indicator variable for missingness was created for each variable with missing data to explore reasons for missingness using logistic regression. Since the outcome variable was binary (died or survived), multiple imputation by chained equations (MICE) was used as a method of choice for imputing missing data, and models containing imputed datasets were compared to the complete-case analysis model. As the variables included in our analysis had very small proportion of missing data, complete-case analysis was retained as our final model.”

Results: in the first paragraph, what do you mean by "accessed a health facility"? Accessed ANC, gave birth at a facility, or something else?

Response: Thank you for the comment. We have explained the variable 'accessing a health facility' in the methods section of the revised draft.

The maternal mortality ratio paragraph doesn't seem to add anything more than what is already available in Figure 1. Consider revising.

Response: We agree with the reviewer and have revised the paragraph.

Also, consider separating the "overall" columns in Figure 1 or at least putting them at one end or the other, rather than in between the various states.

Response: We have revised Figure-1 as suggested by moving the 'Overall' column to the end of the graph.

Can you include a qualitative interpretation of the results of your sensitivity analyses? They do not seem to be mentioned again in text after the methods.

Response: The second last paragraph in the results section describes the findings of the sensitivity analysis for both missing data and clustering. We have further clarified the results in the paragraph.

Discussion: please choose one name for the area under the curve and use it throughout; there are several inconsistent names used.

Response: Thank you for pointing out the inconsistency. We have carefully edited this in the revised draft.

Page 15, line 51, I would advise moving away from a description like "demonstrates convincingly". Perhaps "provides evidence" is more appropriate.

Response: We have revised the sentence as advised. Thank you again for the comments.

Reviewer: 2

Reviewer Name

Monica Saucedo

Institution and Country

Inserm U1153, Obstetrical, Perinatal and Pediatric Epidemiology Research Team. France

Please state any competing interests or state 'None declared':

Non declared

Please leave your comments for the authors below Thanks for give the opportunity to review this interesting paper.

This study estimate the maternal mortality ratio and identifies the risk factors for maternal mortality in nine socioeconomically disadvantaged states in India, a country with the highest RMM, using information from the Indian Annual Health Survey (2010-2013). The study population included more than 7000 maternal deaths among >1.9 million pregnant women. The results should be of interest to readers of BMJ Open in particular from LIMIC.

Response: We thank the reviewer for the comments and suggestions.

The article is well written and the results are clearly presented I have few comments and suggestions for authors :

In the abstract, please give the 95% CI for MMR, as well as in the text P5, L13, it will be useful for the readers to know the recorded information regarding maternal deaths in the AHS questionnaire, in the supplementary material.

Response: We have added the 95% CI for the calculated MMR in the abstract and the results section, as well as in Figure-1. Unfortunately, the AHS questionnaire is not available in the public domain. The dataset comes only with a list of variables and their description. Thus, we are unable to include the questionnaire as supplementary material.

Also in the methods, please give more details about documentation of maternal deaths, the data from AHS for maternal deaths is comparable to a "verbal autopsy" ?

Response: We thank the reviewer for the suggestion. We have added the following paragraph in the revised draft –

"The maternal death questionnaire gathered information from a member of the household in which a woman had died in pregnancy or postpartum during the reference period. The questionnaire focused on identified delays to care and factors contributing to the death, but the methodology was not comparable to the conventional verbal autopsy."

Are data about time of death available? During pregnancy, during labor and delivery or in the puerperium?

Response: The data on time period of death in relation to pregnancy is available. We initially did not include the Figure that showed the results of the analysis of time-period of death, which is now included in the revised draft (Figure 3)

P5L40: It is confusing the study population figure (S1), the exclusion from abortion. In the figure it is stated "abortion related deaths " n=94 517. Please give the n for survivors and for pregnancy related deaths.

Response: We regret that Figure S1 appeared confusing, which we think was mainly because we did not explain why the abortion-related pregnancy deaths were shown to be excluded from the analysis of 'risk factors for maternal mortality'. We have also revised Figure-S1 to indicate that all women, N=94,517, who had an abortion in the reported index pregnancy (whether survived or died) were excluded from the analysis.

P6L49, identified delays to care appear only in the statistical analysis, could you please describe them early in the methods indicating how they were assessed.

Response: As suggested by the reviewer, we have added the following in the methods section of the revised draft –

"The information on 'delay' in the dataset were mainly categorised into 'delay on the part of the woman or the family in recognising the danger signs or seriousness of the complication', 'delay in accessing healthcare', or 'delay in receiving appropriate healthcare'."

P9 L3, In the groups expressing a delay in care seeking or in reaching the facility, over 80% had not registered the death. Is this mean that there wasn't a death certificate for them? Where the deaths are registered in general?

Response: Registered or unregistered deaths were defined based on whether the death was registered with the local registrar for vital statistics within the administrative unit (irrespective of

whether a death certificate was issued or not). We have explained this in the methods section of the revised paper under 'Statistical analysis'.

Reviewer: 3

Reviewer Name

Santiago Garcia-Tizon Larroca

Institution and Country

Hospital General Universitario Gregorio Marañón de Madrid. Servicio de Ginecología y Obstetricia. Madrid. Spain

Please state any competing interests or state 'None declared':

None declared

Please leave your comments for the authors below

First of all I would like to thank you for the opportunity to review this piece of work from the authors as a great effort to address and analyze relevant risk factors related to maternal mortality in such a big sample of patients. This is an outstanding study and I hope I can give you some ideas that might boost your manuscript for publication if you consider them suitable for your research.

Response: We thank the reviewer for the comments and suggestions.

1-Article summary/Abstract/Strengths and limitations of this study

In the last point you mention that "the survey design likely introduced...under-reporting of maternal deaths in early pregnancy and deaths due to indirect causes"

I see that this is also described in the DISCUSSION section (page 17, line 52) but the reasons why this happened are not explained. Is it possible for you to report this issue? Is it documented elsewhere? If it is not possible you should say it. It is positive that you mentioned it though.

Response: We have explained this point further in the discussion section. Literature where this was reported previously was already cited, but we have now provided further reasons. The paragraph has been revised as follows –

"The survey design is also likely to have introduced non-systematic reporting bias. In keeping with previous literature, underreporting of early pregnancy deaths, particularly those arising from complications of abortion or deaths prior to detecting pregnancy, and of maternal deaths arising from indirect causes that might have been reported as deaths due to medical causes (ignoring the pregnancy state) could have led to an underestimation of maternal deaths²⁵."

2-Introduction

Page 4, line 4. You mention that the burden of maternal deaths is reflected by other socio-economic indicators such as poverty and education. Is it possible for you to report more bibliography related to this? Maybe other social determinants of health such as family income, poverty index of the maternal country of origin or human development index of the maternal country of origin that have been related to this complication?

Response: We thank the reviewer for the comment. We were unable to find studies that specifically looked at the association between maternal mortality and poverty or human development index for each state in India. We have however, cited two papers that have looked at the state-wise association of maternal mortality with poverty and education in the country, as well as variation in access to

pregnancy and postnatal care across the northern and southern states. We have revised the sentence as follows –

“This inequality in the burden of maternal deaths between northern and southern states is reflected by other socio-economic indicators such as poverty and education⁷ which could determine access to care during pregnancy and postpartum⁸.”

3-Methods/Variable choice and extraction

Page 6, line 16. You use a classification based on a theoretical framework where you divide factors into 3 groups: Distal, intermediate and proximal. Is this a classification created by you? Is it the WHO theoretical framework? I see the WHO report “The Social Determinants of Maternal Health. Adapted from WHO (2011) Closing the Gap: Policy into Practice on Social Determinants of Health. Discussion for the World Conference on Social Determinants of Health. Geneva: World Health Organization” (https://apps.who.int/iris/bitstream/handle/10665/44731/9789241502405_eng.pdf?sequence=1&isAllowed=y) where intermediate factors are described but I'm not sure you are referring to this. You should report where you got this way of classification from or if you just made it yourself, which is perfectly ok. Anyway I understand it is a very clear way to present it also with figure S2 so as a reader I appreciate it.

Response: We thank the reviewer for pointing out the omission on our part. The theoretical framework is inspired by the WHO conceptual framework of the social determinants of health. We have added this in the revised draft and have cited the report mentioned by the reviewer.

4-Tables S1, S2 and S3

You describe proximal factors as YES or NO but you don't specify causes of maternal death or the main medical conditions related to maternal comorbidity. I don't see it in the RESULTS section either. I understand that this is not the purpose of the study but I wonder if it will be published in the future or it was already done previously as a very valuable information for clinicians like me. For instance the main clinical cause of maternal death is haemorrhage throughout the world. Numbers of this complication are quite different between populations regarding social determinants.

Response: The reviewer has raised an important point. We agree that the examination of the individual complications and their association with maternal mortality would provide more information. While this was done as a sub-group analysis for the study, we were not convinced that the self-reported or family-reported complications (for women who died) were a true reflection of the diagnosed complication/s. We felt that there was a high risk of misclassification due to potential reporting errors or biases which could lead to incorrect conclusions from the study. However, we have included the results from the analysis in a Table at the end of this document. If the reviewer and the Editor would like us to add this information, we will be happy to.

Congratulations again for your work again, please take my comments more as suggestions than corrections.

Response: We thank the reviewer again for the valuable suggestions.

VERSION 2 – REVIEW

REVIEWER	Elizabeth Anderson International Center for Research on Women U.S.
REVIEW RETURNED	29-Jun-2020
GENERAL COMMENTS	Thanks for the chance to review the modifications your team made. No additional comments.